# LLM KNOWLEDGE IS BRITTLE: TRUTHFULNESS REPRESENTATIONS RELY ON SUPERFICIAL RESEMBLANCE

## ABSTRACT

For Large Language Models (LLMs) to be reliable, they must learn robust knowledge that can be generally applied in diverse settings—often unlike those seen during training. Yet, extensive research has shown that LLM performance can be brittle, with models exhibiting excessive sensitivity to trivial input variations. In this work, we explore whether this brittleness is a direct result of unstable internal knowledge representations. To explore this question, we build on previous work showing that LLM representations encode statement truthfulness—i.e., true, factual statements can be easily separated from false, inaccurate ones. Specifically, we test the robustness of learned knowledge by evaluating representation separability on samples that have undergone superficial transformations to drive them out-of-distribution (OOD), such as typos or reformulations. By applying semantically-preserving perturbations, we study how separability degrades as statements become more OOD, across four LLM families, five evaluation datasets, and three knowledge probing methods. Our results reveal that internal representations of statement truthfulness collapse as the samples' presentations become less similar to those seen during pre-training. While LLMs can often distinguish between true and false statements when they closely resemble the pre-training data, this ability is highly dependent on the statement's exact surface form. These findings offer a possible explanation for brittle benchmark performance: LLMs may learn shallow, non-robust knowledge representations that allow for only limited generalizability. Our work presents a fundamental challenge for the utility of truthfulness probes, and more broadly, calls for further research on improving the robustness of learned knowledge representations.

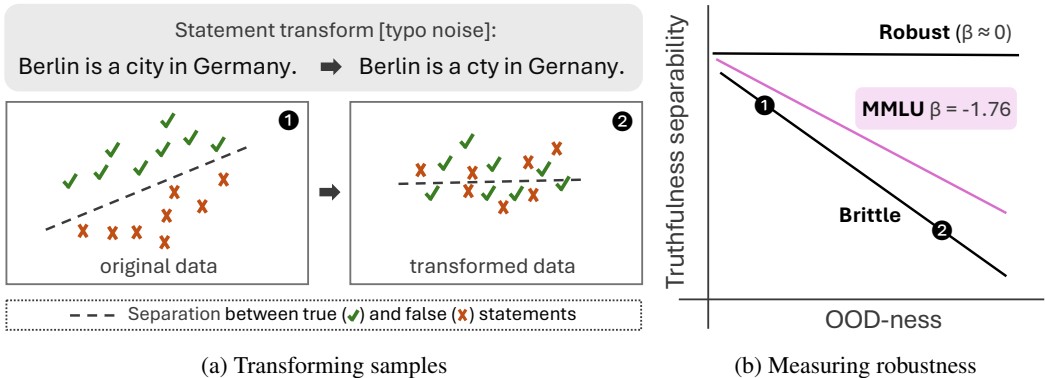

(a) Transforming samples  (b) Measuring robustness

Figure 1: **LLM truth representations degrade under superficial changes.** **(a)** We apply semantically-preserving transformations to shift statements OOD, collapsing the representations of **true** and **false** statements. **(b)** We quantify robustness as the relationship between separability and OOD-ness. A hypothetical **Robust** model would have constant separability regardless of OOD-ness, while a hypothetical **Brittle** model would rapidly degrade. On MMLU, we observe that knowledge representations degrade with increasing perplexity.

# 1 INTRODUCTION

It is expected that LLMs acquire robust and general knowledge during pre-training, which is a critical factor in enabling strong and reliable downstream capabilities. Contemporary LLMs, however, appear to be strikingly brittle, with task performance and benchmark results being highly susceptible to subtle prompt changes and other irrelevant perturbations (Mizrahi et al., 2024; Sclar et al., 2024; Voronov et al., 2024). This brittleness raises a central question: do LLMs learn robust underlying knowledge *representations*, or are they fragile and tied to the exact phrasings and formulations seen during training?

One approach to testing knowledge is via a "true or false" test: correctly determining whether a statement is true requires accurate knowledge of the world. Previous work has suggested that LLM representations encode statement truthfulness, such that representations of true statements are separable from those of false (Azaria & Mitchell, 2023; Li et al., 2023; Burns et al., 2022; Marks & Tegmark, 2023; Wang et al., 2025, *inter alia*). We build on this line of work in order to explore the *generality* and *robustness* of knowledge representations when statements are reformulated or perturbed. Specifically, we explore the extent to which LLMs have a robust internal representation of truthfulness as statements take on presentations increasingly dissimilar to those seen during pre-training, i.e., that are increasingly OOD. For an LLM to have robust knowledge, its knowledge representations should generalize and be robust to superficial changes.

Our method has three components. First, we leverage average perplexity as a proxy measure for how OOD (with respect to pre-training) a given statement is. Second, we apply a range of semantically-preserving perturbations that drive samples OOD, without changing statement meanings. Finally, we test the separability of learned knowledge representations on the perturbed samples, using three different probing techniques. Our analysis of over $2,000$ probes allows us to directly assess how separability degrades as a function of OOD-ness, across four model families and four datasets.

Our findings reveal that, strikingly, the robustness of LLMs' internal knowledge representations is tightly coupled to superficial resemblance to pre-training data. We find that truthfulness separability degrades as samples become increasingly OOD, across all tested truthfulness probing techniques, and regardless of model choice or dataset. Comparing different areas of knowledge, we note that some topics (e.g. marketing and sociology) appear to be less susceptible to distribution shifts than others (e.g. history), indicating they are learned more robustly. Surprisingly, topic robustness is not explained by topic representation in the pre-training data, suggesting that *scaling up pre-training corpora may be insufficient for encoding robust and general knowledge.*

In summary, our results suggest that LLMs learn shallow, non-robust knowledge representations that fail to generalize to scenarios unlike those seen during training. While previous work has highlighted limited generalization performance of probing methods (Wang et al., 2025; Azizian et al., 2025; Orgad et al., 2024; Beigi et al., 2024; Levinstein & Herrmann, 2025), our work demonstrates that this problem stems from brittle internal representations, presenting a significant challenge for probing-based approaches to improving factuality or reliability. More broadly, our work points to a deeper limitation in how LLM knowledge is encoded and applied: rather than developing stable, generalizable representations, current LLMs appear to rely on brittle surface features, undermining their reliability in common contexts.

# 2 RELATED WORK

**LLM robustness.**     Though state-of-the-art LLMs continue to demonstrate increasing scores on popular benchmarks, a large body of research suggests this performance may be unusually brittle. Previous work has found that LLMs are highly sensitive to minor formatting variations (Gu et al., 2023; Sclar et al., 2024; Habba et al., 2025), semantically-equivalent paraphrases (Mizrahi et al., 2024; Sun et al., 2024) and human-validated adversarial perturbations (Wang et al., 2021). In multiple-choice question answering, LLMs are sensitive to option ordering (Pezeshkpour & Hruschka, 2024; Gupta et al., 2024; Alzahrani et al., 2024) and presentation (Alzahrani et al., 2024). In few-shot settings, example formatting and ordering can lead to performance degradations (Zhao et al., 2021; Turpin et al., 2023). While the causes of brittleness may vary by task (Chatterjee et al., 2024) and specific prompt features (Leidinger et al., 2023), these trivial variations suffice to significantly re-order benchmark model rankings (Alzahrani et al., 2024; Mizrahi et al., 2024). Exploring

robustness beyond benchmarking, recent works have also found that simple variations of political survey questions suffice to completely alter expressed opinions (Haller et al., 2024; Shu et al., 2024; Ceron et al., 2024).

One possible cause of such brittleness is that LLMs fail to generalize far beyond their training data. Razeghi et al. (2022) find that LLMs exhibit better numerical reasoning performance on prompts that are better represented during pre-training. Using prompt perplexity as a proxy measure of pre-training representation, Gonen et al. (2023) show that higher perplexity—i.e., more OOD— prompts lead to worse performance. Motivated by and building upon this line of research, in this work we attempt to distinguish between brittle *performance* and brittle *knowledge*, by evaluating the robustness of internal representations in increasingly out-of-distribution scenarios.

**Probing knowledge representations.** Extensive prior work has investigated the internal representations of LLMs as a way of understanding what they have learned during training. Azaria & Mitchell (2023) train multi-layer perceptron (MLP) classifiers on hidden layer activations to separate true, factual statements from false, inaccurate statements, suggesting that LLM internal representations encode a notion of statement truthfulness that can be later recovered. Others (Burns et al., 2022; Li et al., 2023; Marks & Tegmark, 2023; Wang et al., 2025) have suggested that these internal representations of true and false statements are *linearly* separable. Slobodkin et al. (2023) find that answerable and unanswerable questions are also linearly separable, while Gottesman & Geva (2024) train linear probes to predict downstream task performance about a given subject.

Using model outputs rather than internal representations, Kadavath et al. (2022) test whether models can, given a question, distinguish between correct and incorrect answers. Kadavath et al.'s token likelihood-based method, "P(True)", can separate true and false question-answer pairs for larger-scale models. In this work, we leverage output-based methods and both linear and non-linear internal probes to explore the robustness of learned knowledge representations.

**Probe generalization.** While Marks & Tegmark (2023) argue that LLMs encode a "geometry of truth" that is consistent across tasks, such that trained probes should generalize to novel settings, other researchers have found a more mixed picture. On the one hand, a significant body of work finds that trained probes for statement truthfulness fail to generalize across datasets and tasks (Wang et al., 2025; Azizian et al., 2025; Beigi et al., 2024; Orgad et al., 2024), and particularly across different domains (Beigi et al., 2024; CH-Wang et al., 2024). Directly relevant to our work, Levinstein & Herrmann (2025) find that Azaria & Mitchell's non-linear truthfulness probes fail to generalize to even negated variants of their training data. On the other hand, truthfulness probes may generalize to tasks requiring similar skills (Orgad et al., 2024) or reasoning strategies (Zhang et al., 2025), while Slobodkin et al. (2023) find partial generalization of answerability probes to different datasets.

Crucially, each of these works explore the generalization of the *trained probe*—i.e., a probe is trained on one dataset before being tested out-of-distribution. While this is a helpful test of the practical utility of probes, testing probe generalization tells us nothing about how generally the LLM can deploy its knowledge. In contrast, our work directly evaluates the generalizability of *internal representations* via testing how separable they remain OOD—i.e., our probes are both trained and tested in out-of-distribution settings. We interpret the poor performance of an OOD-trained probe as evidence of non-robust knowledge representations.

## 3 METHODS

Our goal is to quantify the robustness of knowledge representations as statements become increasingly dissimilar to those seen during pre-training. See Fig. 1 for an overview of our approach. **First**, we evaluate three truthfulness probing methods that separate true statements from false (§3.1) on four model families (§3.2) and four datasets (§3.3). **Second**, we select a diverse set of meaning-preserving transformations to push dataset samples OOD (§3.4). **Third**, we measure how OOD a transformed statement is using statement perplexity as a proxy (§3.5). **Finally**, we evaluate the degradation of truthfulness representation separability as samples become increasingly OOD (§3.6).

## 3.1 Probing techniques

We evaluate the separability of internal representations according to statement truthfulness using three approaches, two activation-based and one output-based. Probe performance is evaluated using Area Under the Receiver Operating Characteristic Curve (AUC). See Appendix A for full details.

**Non-linear activations classifier.** Following Azaria & Mitchell (2023), we train a 3-layer feed-forward neural network to classify the internal representations of statements according to whether they are true or false.[1] Representations are the residual stream activations of the final token in the statement. Again following Azaria & Mitchell (2023), we test activations from multiple layers using six-fold cross-validation, and report performance via AUC on the best-performing layer for each combination of model and probe.

**Linear activations classifier.** In light of prior work suggesting that true and false statements may have linearly separable representations (Li et al., 2023; Burns et al., 2022; Marks & Tegmark, 2023), we implement a linear alternative to the non-linear activations classifier. We follow the same method as above, but replace the 3-layer feedforward neural network with a single-layer linear network, equivalent to logistic regression.

**P(True).** In addition to the activation-based probes, we employ one method based on the output next-token distribution. Kadavath et al. (2022) introduce *P(True)*, where the model is prompted with a multiple-choice question as to whether the given statement is true or false. We employ standard practice for likelihood-based evaluation by extracting the probabilities of the tokens corresponding to true and to false, before normalizing to obtain a probability for the statement being true. Following Kadavath et al., we use a 6-shot approach, providing the model with six in-context examples. See Appendix A for full details and prompt examples.

## 3.2 Models

We probe the knowledge representations of ten decoder-only autoregressive language models across four model families: OLMo 7B Base and Instruct (Groeneveld et al., 2024), OLMo-2 Instruct (7B and 13B; Walsh et al., 2025), Llama 3.1/3.2 Instruct (1B, 3B, 8B and 70B; Llama Team, AI @ Meta, 2024) and Gemma-3 (4B and 12B; Gemma Team, Google DeepMind, 2025). We make use of Hugging Face transformers (Wolf et al., 2020) for extracting activations and vLLM (Kwon et al., 2023) for efficient inference. See Appendix A for full details and model hyperparameters.

## 3.3 Datasets

We assess the robustness of truthfulness representations on four widely used benchmark datasets, which vary in knowledge domains, question formats, and reliance on retrieval vs. reasoning.

**True-False.** The dataset introduced by Azaria & Mitchell (2023) consists of simple true/false statements across six topics such as cities, animals and facts, making it the most retrieval-oriented benchmark in our suite. The questions are syntactically regular and have low variance in phrasing, which makes them particularly well-suited for testing superficial memorization and direct retrieval.

**MMLU.** The Massive Multitask Language Understanding benchmark (MMLU; Hendrycks et al., 2021) is a multiple-choice question-answering dataset covering a wide range of domains such as STEM, humanities, and professional knowledge. Each question has four candidate answers, and the benchmark includes fine-grained category labels that enable by-topic analysis.

**OpenBookQA.** OpenBookQA (Mihaylov et al., 2018) consists of questions that are less trivia-like than those in the True-False dataset, often requiring longer phrases and some degree of reasoning beyond direct retrieval.

**TruthfulQA.** TruthfulQA (Lin et al., 2022) is arguably the most challenging and diverse benchmark in our evaluation. The questions are designed to probe models' susceptibility to common misconceptions and false beliefs, rather than simple retrieval. Each question is annotated with cate-

---

[1] Where "true" and "false" describe the statement's truth value with respect to the world, rather than some notion of concordance with hidden beliefs.

gories (e.g., misconceptions, superstitions), making the dataset particularly useful for understanding which types of knowledge are robust.

**Statement formatting.** For the True-False dataset, we directly use the original statements without modification. For the multiple-choice datasets (MMLU, OpenBookQA, and TruthfulQA), we combine each question with one of its candidate answers to form a complete question–answer pair.

## 3.4 TRANSFORMATIONS

We consider different types of semantically-preserving transformations to break the samples' resemblance to the pre-training data.[2] See Fig. 1 for examples and Appendix A for further detail.

**Typos and punctuation noise.** We introduce character-level perturbations in the form of typos and punctuation noise using the AugLy data augmentation library (Papakipos & Bitton, 2022), evaluating multiple variants with progressively increasing intensity. Such noise-based augmentations allow us to test the extent to which probes and LLMs rely on exact lexical and orthographic cues when separating true from false statements.

**Negation.** We employ the `negate` Python library[3] to generate syntactic negations. These negated sentences remain syntactically well-formed and semantically interpretable, while systematically flipping the truth value of the original statement.

**Yoda speak.** We use the NL-Augmenter library (Dhole et al., 2023) to flip the clause structure such that it reads like "Yoda speak". This transformation typically results in sentences with a non-canonical word order as in the example: "Much to learn, you still have." Because these configurations are both syntactically valid and infrequent in ordinary English text, they are unlikely to be seen during pre-training.

**Translation.** We further drive samples OOD by translating them into French and Spanish using the NLLB-200 machine translation model (Costa-Jussà et al., 2022). This transformation preserves the semantic content but alters virtually all surface-level statistics of the input.

## 3.5 MEASURING OOD-NESS

Each transformation can be thought of as a different distribution shift. Thus, when plotting performance degradation under each transformation, we use OOD-ness as a common scale to evaluate robustness. Given LLM training data is typically unavailable, we are unable to *directly* evaluate whether a statement is OOD, so we instead follow previous work (Razeghi et al., 2022; Gonen et al., 2023) and use statement perplexity as a proxy. Assuming a given LLM is a reasonable model of its training data, it should assign low likelihoods to less well-represented, i.e., more OOD, samples.

Given a token sequence $\mathbf{u} = \langle u_1, \ldots, u_N \rangle$, and a language model $P_\theta$, the perplexity of $\mathbf{u}$ is the exponentiated average negative log-likelihood of the sequence,

$$\mathrm{PPL}(\mathbf{u}) = \exp\left\{ -\frac{1}{N} \sum_{i=1}^{N} \log P_\theta(u_i | \mathbf{u}_{<i}) \right\},$$

where $P_\theta(u_i | \mathbf{u}_{<i})$ is the model's estimated conditional probability of observing token $u_i$ given preceding tokens $\mathbf{u}_{<i}$.

**Validating perplexity as a proxy OOD measure.** To validate that perplexity is an appropriate proxy for measuring how OOD a statement is, we make use of the OLMo model family (Groeneveld et al., 2024) and its publicly-available pre-training data, Dolma (Soldaini et al., 2024). We use the Infini-gram API (Liu et al., 2024) to access the Dolma n-gram statistics for each sample. In Appendix C, we show that OLMo statement perplexities are highly correlated with Dolma n-gram counts, suggesting perplexity can proxy how well-represented a sample is in the pre-training corpus. We additionally reevaluate our main findings using average n-gram counts rather than perplexity, finding that our conclusions remain consistent regardless of approach.

---

[2]To verify that our transformations preserve the original statements' truth value, we conducted a human validation study, provided in Appendix B.

[3]https://pypi.org/project/negate

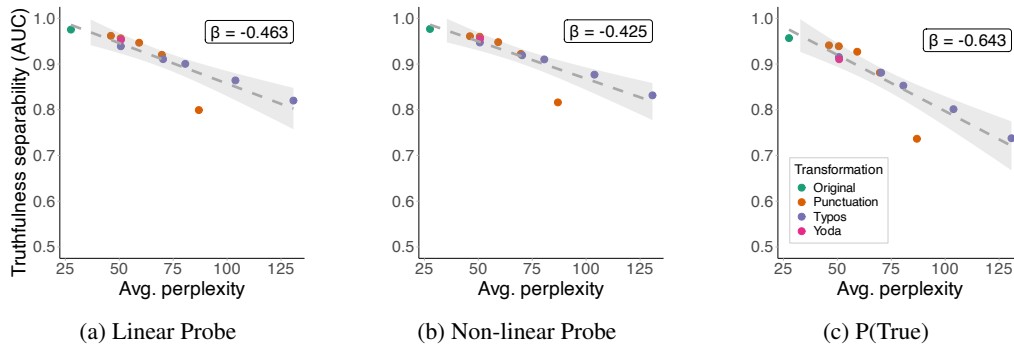

Figure 2: Truthfulness separability (probe AUC) against average perplexity on the True-False dataset for Llama 3.1 8B Instruct for the **(a)** linear, **(b)** non-linear, and **(c)** P(True) probes. **Probe performance degrades as samples become more OOD across all tested probes, suggesting knowledge representations are not robust.**

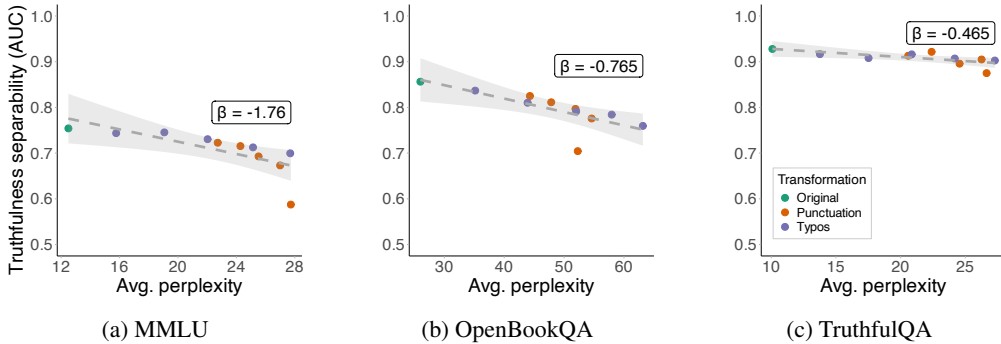

Figure 3: Non-linear probe performance (AUC) against average perplexity for Llama 3.1 8B Instruct on **(a)** MMLU, **(b)** OpenBookQA, and **(c)** TruthfulQA. **Despite differences in AUC on the original dataset (green dots), truthfulness representations consistently degrade on all datasets.**

### 3.6 PUTTING IT ALL TOGETHER: TESTING REPRESENTATION ROBUSTNESS

Given a transformed dataset, we measure the separability of true and false statements using probe AUC, and measure how OOD the transformed dataset is via average statement perplexity. Then, we measure robustness as the standardized slope ($\beta$) of a linear regression model, where the predictor is average perplexity, and the response is probe AUC. A steeper negative slope indicates rapid degradation and less robust representations, while a flatter slope indicates higher robustness.

## 4 TRUTHFULNESS REPRESENTATIONS RELY ON SUPERFICIAL RESEMBLANCE

First, we test the robustness of knowledge representations across different probing methods, datasets, and models. Later, in §5, we explore variability in robustness in greater depth.

**Knowledge representations degrade OOD.** First, we test the robustness of three different probing methods, P(True), the non-linear probe, and the linear probe, as statements become increasingly OOD, as measured by perplexity. In Fig. 2, we show probe AUC as a function of average statement perplexity for Llama 3.1 8B Instruct on the True-False dataset. All three methods achieve high AUC on the original, untransformed True-False dataset (AUC $\geq$ .96), indicating that true and false statements are initially separable. However, AUC consistently degrades as we move out of distribution, with P(True) exhibiting lower robustness ($\beta = -.64$) compared to the linear and non-linear probe ($\beta = -.43$ and $\beta = -.46$, respectively).[4]

---

[4] As a sanity check, we additionally analyze robustness across all model **layers**. See Appendix C.4.

**Representations degrade consistently across datasets.** Next, we test robustness across datasets using the non-linear probe on Llama 3.1 Instruct 8B. Across TruthfulQA, OpenBookQA, MMLU, and True-False, probe performance again begins well above chance on the original data (AUC $\in$ [.75, .98]), though all exhibit degradation under distribution shift (Figure 3). The steepest decline occurs for MMLU ($\beta = -1.76$), followed by OpenBookQA ($\beta = -.77$), True-False ($\beta = -.43$) and finally TruthfulQA ($\beta = -.47$). For results with additional models and probes, see Appendix C.

**Representations degrade consistently across model families and scales.** Finally, we show that truthfulness representations degrade regardless of the LLM choice. Using the non-linear probe, Fig. 4a illustrates that most models exhibit similar rates of degradation, with Llama 3.1 Instruct 70B—the largest model—being the least robust ($\beta = -1.53$), and Gemma-3 4B the most ($\beta = -.07$). For additional probes and datasets, see Appendix C. In Fig. 4b, comparing different Llama 3.1 model scales, we see that the non-linear probe exhibits much sharper degradation for 70B models. In contrast, P(True) shows a slight positive effect of scale on robustness, aligning with Kadavath et al.'s (2022) finding that larger models are better calibrated.

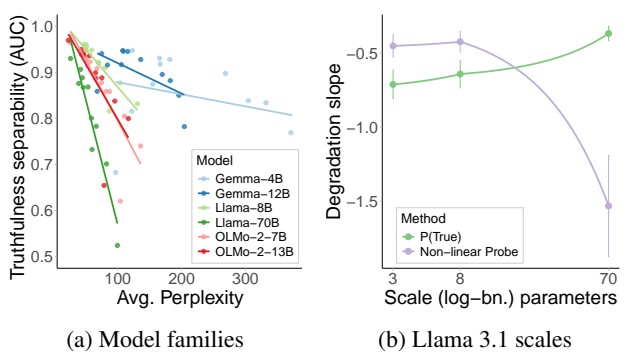

(a) Model families      (b) Llama 3.1 scales

Figure 4: **(a)** Non-linear probe performance (AUC) against average perplexity for various model families. **(b)** Degradation slope for the non-linear and P(True) probes at increasing Llama 3.1 Instruct scales. **All models suffer degraded representations under OOD shift; increasing scale may worsen representation robustness.**

**Summary.** We find that truthfulness representations consistently degrade under superficial modifications, across probing methods, datasets, and models, indicating an over-reliance on the exact phrasings and formulations seen during training.

## 5 NOT ALL REPRESENTATIONS ARE EQUAL

Having explored the robustness of truthfulness representations *in general*, we now ask whether certain types of knowledge are learned more or less robustly.

**Benchmark performance does not imply robust representations.** We first ask whether correctly answering benchmark questions corresponds to more robust representations. To do so, we prepare a filtered subset of MMLU questions where the model responds correctly during bench-

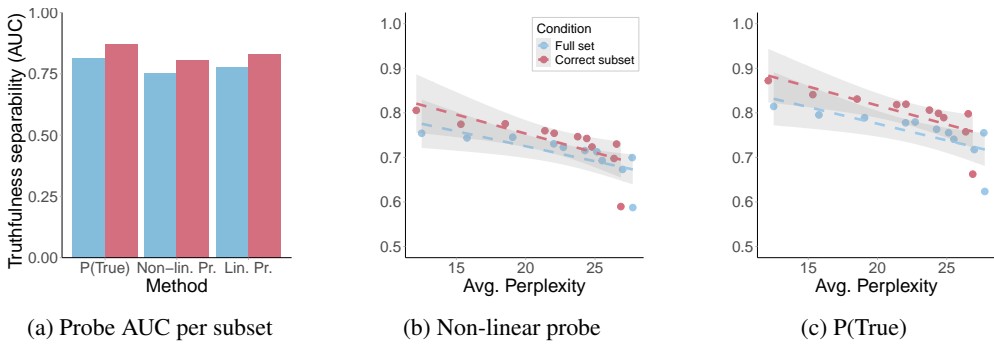

(a) Probe AUC per subset      (b) Non-linear probe      (c) P(True)

Figure 5: **(a)** Probe performance (AUC) for Llama 3.1 8B Instruct on the correct-only MMLU subset (red) and the full dataset (blue). **(b)** Non-linear probe and **(c)** P(True) performance (AUC) against average perplexity for correct and full sets. **Knowledge representations still degrade even when the model responds correctly during benchmarking.**

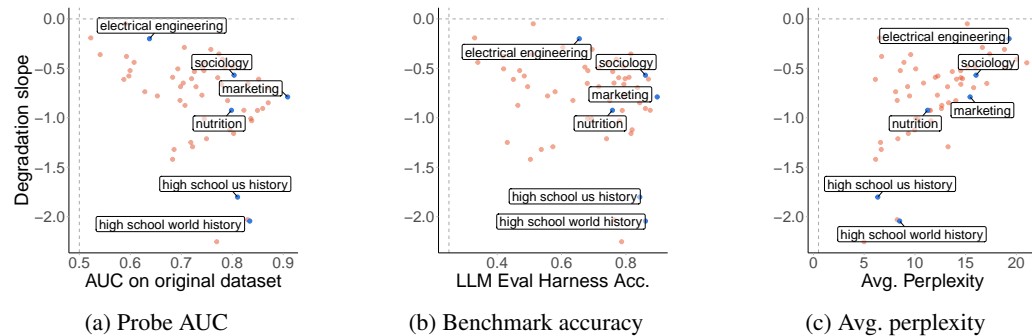

Figure 6: Non-linear probe robustness by MMLU topic against original, unmodified dataset **(a)** probe AUC, **(b)** benchmark accuracy, **(c)** and average perplexity for Llama 3.1 8B Instruct. **Certain topics are robustly separable, even with higher perplexity.**

marking (see Appendix A), before comparing probe AUC on this subset against AUC on the full dataset. As shown in Fig. 5a, AUC scores are consistently higher for the correctly-answered subset.

However, Fig. 5b and Fig. 5c show that for both P(True) and the non-linear probe, the degradation slopes on the full dataset and the correct subset are nearly parallel (P(True): $\beta_{\text{full}} = -.59$ vs. $\beta_{\text{subset}} = -.67$; non-linear: $\beta_{\text{full}} = -.53$ vs. $\beta_{\text{subset}} = -.67$), suggesting that knowledge representations degrade equally whether or not the model can answer the question in a benchmarking setting.

**Certain topics have more robust representations.** Next, we break out our analysis by statement topic, using the non-linear probe on Llama 3.1 8B Instruct, over the full set of MMLU topics (see Appendix A.5). Figure 6a shows the robustness of each MMLU topic (degradation slope) against the initial separability (AUC on the original dataset). Topics in the upper-right quadrant can be considered "well learned", achieving high AUC while degrading only minimally under OOD shifts. Example topics with robust and separable representations include sociology and marketing. By contrast, topics such as high school world history achieve high separability in-distribution, but degrade sharply under minor perturbations. Interestingly, high school world history questions also tend to have relatively long sentences, and we observe that topics with longer sentences sometimes exhibit lower robustness (Fig. S6b).

Figure 6b shows topic robustness (degradation slope) against benchmark accuracy on the original dataset. Even for topics where models achieve high benchmark scores, the robustness varies: certain high-scoring topics (e.g., sociology) display greater robustness than others (e.g., marketing).

We explore whether topics that are more in-distribution are more robust in Fig. 6c, which shows topic robustness (degradation slope) against the topic's average perplexity—again making use of perplexity as a proxy for training data coverage. Interestingly, topics that that begin *more* OOD (higher perplexity) might also be *more* robust to subsequent shifts (e.g., electrical engineering), while better-represented topics can degrade sharply (e.g., high school US history).

**LLMs are more susceptible to certain transformations.** Finally, we examine whether knowledge representations are more susceptible to certain kinds of transformation. Figure 7 shows the change in AUC of each transformed dataset variant against the change in average perplexity, using the non-linear probe with Llama 3.1 8B Instruct on the True-False dataset. Results for the linear probe and P(True) are shown in Fig. S7 for comparison. For the punctuation noise, typo, and Yoda transformations, we see that change in AUC is proportional to

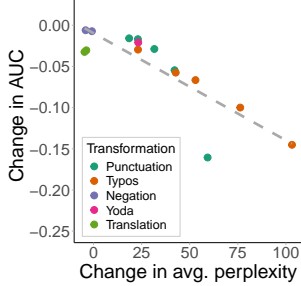

Figure 7: Change in non-linear probe performance (AUC) against change in average perplexity on True-False dataset. **While most transformation types degrade AUC in line with increasing perplexity, translation breaks representations while remaining in-distribution.**

change in perplexity: the further samples move OOD, the greater the degradation.

In contrast, while translation does not measurably increase perplexity, it has a more pronounced effect on AUC. This suggests that knowledge representations may degrade rapidly even when translated samples remain relatively in-distribution.

For the non-linear probe, negation neither shifts samples OOD nor impacts AUC, suggesting representations that are robust to negation. While prior work (Levinstein & Herrmann, 2025) has highlighted limited generalization of non-linear probes to negated samples, our finding suggests this is a problem with the trained probe, rather than the underlying representations.

**Summary.** While truthfulness representations are generally brittle, LLMs may learn certain topics more robustly, and be more robust to certain types of shift. Lower perplexity topics are not necessarily learned more robustly.

# 6 DISCUSSION

We have explored whether LLMs learn robust and generalizable knowledge representations. Our results show that learned truthfulness representations degrade under superficial changes in input presentation across probing methods, model families, and datasets. Even when an LLM typically responds correctly in a standard benchmarking setup, its internal representations are no more robust. While certain subject areas appear to be learned more robustly, this does not appear to be explained by coverage in the pre-training data.

**Connection to benchmark brittleness.** As discussed in §2, LLM benchmark scores can be significantly influenced by trivial changes including paraphrasing and formatting changes (e.g. Gu et al., 2023; Sclar et al., 2024; Habba et al., 2025; Mizrahi et al., 2024; Sun et al., 2024). Yet, it remains unclear whether such brittleness stems from an inability to generalize and apply robustly-learned knowledge when performing a task (i.e. brittle *performance*), or a lack of robustness in the underlying knowledge representations themselves (i.e. brittle *knowledge*). While not ruling out additional factors that may cause brittle performance, our results indicate that brittle knowledge representations are likely to play a key role.

**Eliciting latent knowledge.** During pre-training, it is expected that LLMs jointly learn how to use language and store knowledge about the world (allal et al., 2025). Our finding that benchmark performance need not imply robust representations is in line with previous work suggesting a disconnect between latent knowledge and performance (Li et al., 2023). Previous work has argued that subsequent post-training primarily "elicits" or "sharpens" pre-existing latent knowledge, rather than teaching fundamentally new capabilities (Zhou et al., 2023; Burns et al., 2024; Ye et al., 2025; Muennighoff et al., 2025; Yue et al., 2025). If this is the case, developing robust and generalizable knowledge representations—which our work suggests are lacking—will be of paramount importance.

**Effect of pre-training coverage.** Surprisingly, our experiments on MMLU topics reveal that lower-perplexity topics are not necessarily more robust. Interpreting statement perplexity as a proxy for pre-training coverage (Razeghi et al., 2022; Gonen et al., 2023), this suggests that increasing the number and variability of occurrences does little for robustness, though improved data quality (Li et al., 2025; allal et al., 2025) is an exciting avenue for future research.

**Limitations.** In order to explore representation robustness in settings where statements superficially differ from those seen in training, we make use of a variety of artificial transformations. A key limitation of this approach, however, is that not all transformations are equally naturalistic. While a transformation such as punctuation noise is unlikely to occur during regular user interaction, we are principally interested in it as a tool for driving samples OOD, rather than being a realistic test of deployed behavior. A more naturalistic way of driving statements OOD would be through reformulating statements into less expected or less common linguistic forms. However, reliably generating paraphrases that preserve meaning while ensuring sufficient distributional shift is non-trivial and difficult to automate.

**Conclusion.** We explored the robustness of learned knowledge representations, finding that LLMs rely on superficial resemblance to their training data in order to determine statement truthfulness. In light of our results, an exciting area for future work is in methods for improving the robustness of

knowledge representations. Ultimately, we see improving the generalizability and applicability of learned knowledge as a fruitful path toward more robust and reliable LLMs.

## REPRODUCIBILITY STATEMENT

Our work is fully reproducible: all methods rely on published research, open-weight models (see Appendix A.4), and publicly available datasets (see Appendix A.5). Full details of the linear and non-linear probes, including representation extraction and classifier training, are provided in Appendix A.1.1 alongside code snippets in Appendix D, and details for the P(True) method are given in Appendix A.2. Data preprocessing, perturbation pipelines, and evaluation procedures are described in the main paper and appendix (see §3 and Appendix A), ensuring that all experiments can be readily replicated.

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

## SUPPLEMENTARY MATERIALS OVERVIEW

In the following supplementary materials, we present details of additional methods used in Appendix A, including further details of probe methods, models, datasets, dataset transforms, and an ablation study using n-gram statistics rather than perplexity as an OOD metric. We provide additional supporting evidence in Appendix C, including replications of our main results with different combinations of probing methods, models, and datasets.

## A  ADDITIONAL METHODS

### A.1  PROBE METHODS

#### A.1.1  NON-LINEAR ACTIVATIONS CLASSIFIER

For the non-linear classifier we implement the Statement Accuracy Prediction based on Language Model Activations (SAPLMA; Azaria & Mitchell, 2023): a multi-layer perceptron classifier to predict the truthfulness of a statement from a language model's hidden layer activations. The hidden state of the final token is passed as input to a 3-layer feedforward neural network (256–128–64 hidden units with ReLU activations, sigmoid output), trained for 5 epochs on a balanced dataset of true/false statements using stratified 6-fold cross-validation. Unlike Azaria & Mitchell (2023), we do not treat separate topics as folds for cross-validation, instead concatenating across topics and drawing each fold via uniform sampling. We trained the probe with the Adam optimizer using a learning rate of $1e^{-2}$, without weight decay or learning rate scheduling. Following Azaria & Mitchell (2023), we test six layers (layers 16, 20, 24, 28, -8, and -1) and report results from the best-performing layer for each model/probe combination. For the smaller Llama 3.2-1B model, we test four layers (layers 4, 8, 12, -1). Note that for each dataset, we determine the best layer according to its original (i.e., non-transformed) samples and use this layer to extract the activations of the transformed counter-parts.

### A.2  LINEAR ACTIVATIONS CLASSIFIER

Our approach to the linear activations classifier is identical to the non-linear classifier, but the MLP is replaced with a single-layer linear network. We follow the same cross-validation, layer selection, and optimization procedure as with the non-linear classifier.

### A.3  P(TRUE)

We deploy P(True) (Kadavath et al., 2022) as a test of output-based probing. We prompt the model using a template $T(\cdot)$ that wraps a statement in prompt Fig. S1a which asks the model whether the statement is correct or incorrect in a multiple-choice setup.

Given a statement $\mathbf{u}$, we extract the probability mass the model assigns to the letter A, corresponding to correct, $p_A = P(\text{A}|T(\mathbf{u}))$ and to letter B, corresponding to incorrect, $p_B = P(\text{B}|T(\mathbf{u}))$, and then compute the final score as the normalized probability $\widetilde{p}_A = \frac{p_A}{p_A + p_B}$. Following Kadavath et al. (2022), we use a 6-shot approach, providing the model with six in-context examples.

As shown in Fig. S1a, we use a different prompt for statement-based datasets (e.g., True-False) than for question-answer datasets (e.g., MMLU).

### A.4  MODEL DETAILS

See Table S1 for all models evaluated and corresponding Hugging Face links. We use vLLM (Kwon et al., 2023) for extracting statement perplexity and P(True) token probabilities. Internal representations were extracted using Hugging Face transformers (Wolf et al., 2020). For the comparison against benchmark performance in §5, we use the LM Evaluation Harness (Sutawika et al., 2025) with default settings unchanged. Llama 3.1 70B Instruct was tested at 16-bit precision. Our experiments were conducted on a private compute cluster using NVIDIA Tesla V100 GPUs.

```
Statement: <statement>
Is the above statement
(A) correct
(B) incorrect
The statement is (
```

```
Question: <question>
Response: <response>
Is the above response
(A) correct
(B) incorrect
The response is (
```

(a) Statement prompt

(b) QA prompt

Fig. S1: Prompt templates used for P(True). Statements (True-False dataset) were wrapped in template (a), the QA pairs (remaining datasets) were wrapped in template (b).

| Model | Hugging Face ID |
|---|---|
| Gemma 3 4B Instruct | `google/gemma-3-4b-it` |
| Gemma 3 12B Instruct | `google/gemma-3-12b-it` |
| Llama 3.1 8B Instruct | `meta-llama/Llama-3.1-8B-Instruct` |
| Llama 3.1 70B Instruct | `meta-llama/Llama-3.1-70B-Instruct` |
| Llama 3.2 1B Instruct | `meta-llama/Llama-3.2-1B-Instruct` |
| Llama 3.2 3B Instruct | `meta-llama/Llama-3.2-3B-Instruct` |
| OLMo 7B Base | `allenai/OLMo-7B-hf` |
| OLMo 7B Instruct | `allenai/OLMo-7B-Instruct-hf` |
| OLMo 2 7B Instruct | `allenai/OLMo-2-1124-7B-Instruct` |
| OLMo 2 13B Instruct | `allenai/OLMo-2-1124-13B-Instruct` |

Table S1: List of models evaluated.

## A.5 DATASET DETAILS

Note that for the MMLU results reported in §4, we use a representative subset of 11 topics for computational efficiency as well as to ensure comparable sample sizes with respect to the other tested datasets. The subset consists of: anatomy, business ethics, clinical knowledge, global facts, high-school European history, high-school geography, high-school government and politics, high-school US history, high-school world-history, pre-history, and public relations. For the by-topic analyses based solely on Llama 3.1 8B and the non-linear probe presented in §5, we use the full set of topics.

## A.6 TRANSFORM DETAILS

For the typo transform, we use AugLy (Papakipos & Bitton, 2022) to insert between 1 and 5 character substitutions, deletions, or insertions at random positions.

For the punctuation noise transform, we insert spurious punctuation symbols every 25, 20, 15, 10, or 5 characters, respectively.

## A.7 MEASURING OOD WITH N-GRAM STATISTICS

As we discuss in §3.5, our main results rely on statement perplexity as a proxy measure for how OOD a given statement is, following prior work (Razeghi et al., 2022; Gonen et al., 2023). To validate this approach, we also directly inspect the pre-training data of an open weights language model, OLMo 2 (Groeneveld et al., 2024). We use the infini-gram API (Liu et al., 2024)[5] to obtain the n-gram counts, specifically, 6-gram counts, of all samples, using DOLMA (Soldaini et al., 2024) and OLMo-2 13B (Walsh et al., 2025) as the reference dataset.

Using these n-gram counts, we test an alternative measure of OOD-ness based on the frequency of n-gram occurrences. *Log-average n-gram count* is an extension of metrics such as token and n-gram match, both commonly employed to measure contamination (Singh et al., 2024; Brown et al., 2020).

---

[5]`https://infini-gram.readthedocs.io/en/latest/api.html`

Log-average n-gram count quantifies the *density* with which n-grams from the pre-training corpus appear in the evaluation sequence. Concretely, for each n-gram in the evaluation sequence, we count the number of times that n-gram appears in the pre-training corpus, average these counts over all n-grams in the sequence, and then take the logarithm of this average.

Formally, for a sequence $s$ with $m$ n-grams $g_1, g_2, \ldots, g_m$, and where $c(g_i)$ is the count of $g_i$ in the pre-training corpus, the score is defined as:

$$\text{log-avg-ngram-count}(s) = \log\left(\frac{1}{m}\sum_{i=1}^{m} c(g_i)\right).$$

This metric captures not only whether n-grams appear in the pre-training corpus, but also *how frequently* they appear. We hypothesize that this degree of representation—i.e., how densely a sequence is represented in the pre-training data—is relevant for this work.

In Appendix C we demonstrate that statement perplexity is highly correlated with log average n-gram counts, supporting its use as a proxy for OOD-ness.

### A.8 MMLU BENCHMARKING

For the exploration of the connection benchmarking performance and robustness in §5, we follow the standard MMLU benchmarking setup, running a likelihood-based evaluation using Language Model Evaluation Harness (Sutawika et al., 2025) to extract predicted multiple-choice responses for each question. Then, we use these predictions to prepare a filtered subset of questions where the model responds correctly.

## B HUMAN VALIDATION STUDY

To verify that our transformations preserve truth value from a human perspective, we conducted a small-scale human validation study. The goal of this experiment was to test whether human annotators assign the same truth value to the original version of a statement and to its transformed counterpart. If the transformations altered meaning or introduced semantic drift, we would expect a measurable drop in agreement between these two conditions.

**Setup.** We constructed a set of 90 items, each consisting of (i) an original statement and (ii) one transformed version. We also included original–original pairs as a baseline to estimate natural annotation variability. We considered three types of transformations: Yoda-style syntactic reordering, punctuation noise, and typos. Four annotators participated in the study. Each annotator saw only one version of any given item (either the original or the transformed version), avoiding priming effects. We used a Latin square design with two counterbalanced lists to ensure that across annotators, each item was evaluated exactly once in its original form and once in its transformed form.

Annotators were asked to assign a binary truth value (true/false) to each statement.

**Analysis.** For each item, we compared the truthfulness judgments given by annotators who saw the original version and those who saw the corresponding transformed version. We computed the proportion of items for which annotators agreed in truth value across versions. These original–transformed agreement rates were then compared to the baseline agreement rates between annotators who saw two independent original versions (original–original pairs).

**Results.** Across all transformation types, the inter-annotator agreement between original–transformed pairs ranged from 0.78 to 0.80, which is effectively identical to the original–original baseline. This indicates that the transformations do not systematically alter the perceived truth value of the statements for humans.

**Conclusion.** The human validation study confirms that our transformations are semantically preserving with respect to truth value. Therefore, the degradation in truthfulness separability observed in our probing experiments cannot be attributed to semantic changes introduced by the transformations.

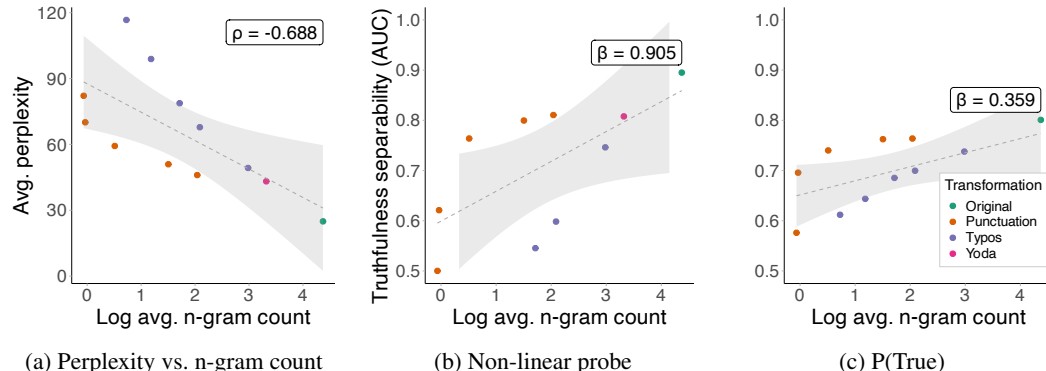

(a) Perplexity vs. n-gram count     (b) Non-linear probe     (c) P(True)

Fig. S2: Comparison of two OOD metrics. We find a strong correlation between the model-free log average n-gram count, based on DOLMA and OLMo-7B Instruct average statement-perplexity (a). Furthermore, representations of truthfulness within the True-False dataset for OLMo-7B Instruct degrade consistently as samples become more OOD (low log-average n-gram count) under various transformations.

## C  ADDITIONAL RESULTS

### C.1  PERPLEXITY IS A GOOD PROXY FOR OOD-NESS

As laid out in §3.5, estimating log-average n-gram counts requires direct access to pre-training data. For our fleet of models, this is only feasible for models of the OLMo family. In order to test the degree to which perplexity and log-average n-gram counts are interchangeable as measures of out-of-distribution, we first assess the correlation between the log-average n-gram count measure, measured with respect to DOLMA (Soldaini et al., 2024) as the reference dataset, and the average statement perplexity, based on OLMo 7B Instruct, for each transform of the True-False dataset.

Fig. S2a illustrates that there is a strong negative correlation between log-average n-gram count and average statement perplexity ($\rho[\mathrm{df} = 10] = -.69, p < .05$), suggesting that statements with dense representations in the pre-training data tend to exhibit low average perplexity.

While this result is a good indicator that the model-based measure and the direct, n-gram based appear to capture the same aspect of out-of-distribution measures, we explore whether the estimated degree of degradation using log-average n-gram count approximates the values obtained in the above experiments using statement perplexity. Figs. S2b and S2c show that truthfulness representations degrade consistently, both for P(True) ($\beta = .36$) and the non-linear probe ($\beta = .91$).

In comparison to the analogous analysis using average statement perplexity as the OOD measure, the absolute standardized slopes are higher here ($\beta = -.64$ for P(True) and $\beta = -.46$ for the non-linear probe), suggesting that the degradation pattern is more pronounced when using log-average n-gram counts.

### C.2  DEGRADATION SLOPES ACROSS DATASETS, MODELS AND PROBES

In the following, we report the magnitudes of degradation slopes across datasets and models in Fig. S3 and in tabular format in Table S2, Table S3, and Table S4. Finally, we show the degradation slopes as scatter plots for a subset of models and for all three probing methods in Fig. S4.

### C.3  EFFECT OF MODEL SCALE

In the analysis of the effect of model scale in §4 we have excluded the smallest Llama model, Llama 3.2 1B Instruct, because the original AUC on the untransformed data is substantially lower (AUC=0.59) compared to its larger counterparts (.94 for 3B, .96 for 8B and .98 for 70B, respectively). Because it starts from a lower AUC, its degradation slope would necessarily be less nega-

| Model name | Slope magnitude ($\beta$) | | | |
| --- | --- | --- | --- | --- |
| | True-False | TruthfulQA | OpenBookQA | MMLU |
| GEMMA-3-12B-Instruct | -0.17 ± 0.07 | -0.44 ± 0.14 | -0.17 ± 0.21 | 0.08 ± 0.68 |
| GEMMA-3-4B-Instruct | -0.07 ± 0.07 | -0.10 ± 0.19 | -0.05 ± 0.08 | 0.29 ± 0.29 |
| Llama-3.1-70B-Instruct | -1.53 ± 0.34 | -2.22 ± 0.92 | -1.77 ± 0.75 | -5.30 ± 1.54 |
| Llama-3.1-8B-Instruct | -0.43 ± 0.07 | -0.46 ± 0.16 | -0.77 ± 0.23 | -1.76 ± 0.56 |
| Llama-3.2-1B-Instruct | -0.71 ± 0.06 | -0.78 ± 0.13 | -0.44 ± 0.05 | -0.47 ± 0.06 |
| Llama-3.2-3B-Instruct | -0.45 ± 0.08 | -0.75 ± 0.17 | -0.73 ± 0.15 | -1.30 ± 0.28 |
| OLMo-2-13B-Instruct | -0.62 ± 0.20 | -0.63 ± 0.29 | -0.50 ± 0.30 | -1.20 ± 0.85 |
| OLMo-2-7B-Instruct | -0.68 ± 0.15 | -0.54 ± 0.17 | -0.66 ± 0.23 | -1.66 ± 0.49 |
| OLMo-7B-Base | -1.45 ± 0.18 | -0.45 ± 0.13 | -0.47 ± 0.15 | -0.40 ± 0.17 |
| OLMo-7B-Instruct | -1.37 ± 0.17 | -0.38 ± 0.09 | -0.50 ± 0.19 | -0.49 ± 0.24 |

Table S2: Magnitudes of regression line slopes for non-linear probe.

| Model name | Slope magnitude ($\beta$) | | | |
| --- | --- | --- | --- | --- |
| | True-False | TruthfulQA | OpenBookQA | MMLU |
| GEMMA-3-12B-Instruct | -0.11 ± 0.08 | 0.00 ± 0.00 | -0.00 ± 0.00 | 0.00 ± 0.00 |
| GEMMA-3-4B-Instruct | 0.00 ± 0.00 | 0.00 ± 0.00 | -0.00 ± 0.00 | -0.00 ± 0.00 |
| Llama-3.1-70B-Instruct | -0.98 ± 0.26 | -2.14 ± 0.95 | -1.42 ± 0.46 | -3.35 ± 1.17 |
| Llama-3.1-8B-Instruct | -0.46 ± 0.08 | -0.42 ± 0.15 | -0.87 ± 0.25 | -1.89 ± 0.51 |
| Llama-3.2-1B-Instruct | -0.47 ± 0.05 | -0.78 ± 0.14 | -0.40 ± 0.03 | -0.46 ± 0.06 |
| Llama-3.2-3B-Instruct | -0.49 ± 0.07 | -0.64 ± 0.15 | -0.77 ± 0.13 | -1.13 ± 0.21 |
| OLMo-2-13B-Instruct | -0.69 ± 0.15 | -0.52 ± 0.28 | -0.56 ± 0.32 | -1.20 ± 0.70 |
| OLMo-2-7B-Instruct | -0.69 ± 0.11 | -0.47 ± 0.18 | -0.68 ± 0.20 | -1.39 ± 0.27 |
| OLMo-7B-Base | -0.79 ± 0.14 | -0.36 ± 0.11 | -0.49 ± 0.18 | -0.23 ± 0.16 |
| OLMo-7B-Instruct | -0.84 ± 0.14 | -0.30 ± 0.08 | -0.49 ± 0.20 | -0.66 ± 0.34 |

Table S3: Magnitudes of regression line slopes for linear probe.

| Model name | Slope magnitude ($\beta$) | | | |
| --- | --- | --- | --- | --- |
| | True-False | TruthfulQA | OpenBookQA | MMLU |
| GEMMA-3-12B-Instruct | -0.19 ± 0.08 | -0.41 ± 0.28 | -0.20 ± 0.12 | 0.04 ± 0.27 |
| GEMMA-3-4B-Instruct | -0.08 ± 0.06 | 0.22 ± 0.36 | 0.04 ± 0.11 | 0.45 ± 0.32 |
| Llama-3.1-70B-Instruct | -0.37 ± 0.05 | -1.36 ± 0.15 | -0.73 ± 0.15 | -0.92 ± 0.35 |
| Llama-3.1-8B-Instruct | -0.64 ± 0.09 | -1.25 ± 0.51 | -0.78 ± 0.23 | -1.95 ± 0.61 |
| Llama-3.2-1B-Instruct | -0.12 ± 0.01 | -0.17 ± 0.15 | -0.23 ± 0.02 | -0.13 ± 0.04 |
| Llama-3.2-3B-Instruct | -0.71 ± 0.10 | -1.29 ± 0.42 | -0.79 ± 0.13 | -1.76 ± 0.30 |
| OLMo-2-13B-Instruct | -0.70 ± 0.13 | -0.58 ± 0.32 | -0.43 ± 0.34 | -1.08 ± 0.72 |
| OLMo-2-7B-Instruct | -0.70 ± 0.11 | -1.20 ± 0.27 | -0.76 ± 0.31 | -1.46 ± 0.41 |
| OLMo-7B-Base | -0.07 ± 0.01 | 0.13 ± 0.13 | -0.10 ± 0.05 | -0.14 ± 0.05 |
| OLMo-7B-Instruct | -0.54 ± 0.13 | -0.11 ± 0.10 | -0.19 ± 0.08 | -0.53 ± 0.21 |

Table S4: Magnitudes of regression line slopes for P(True).

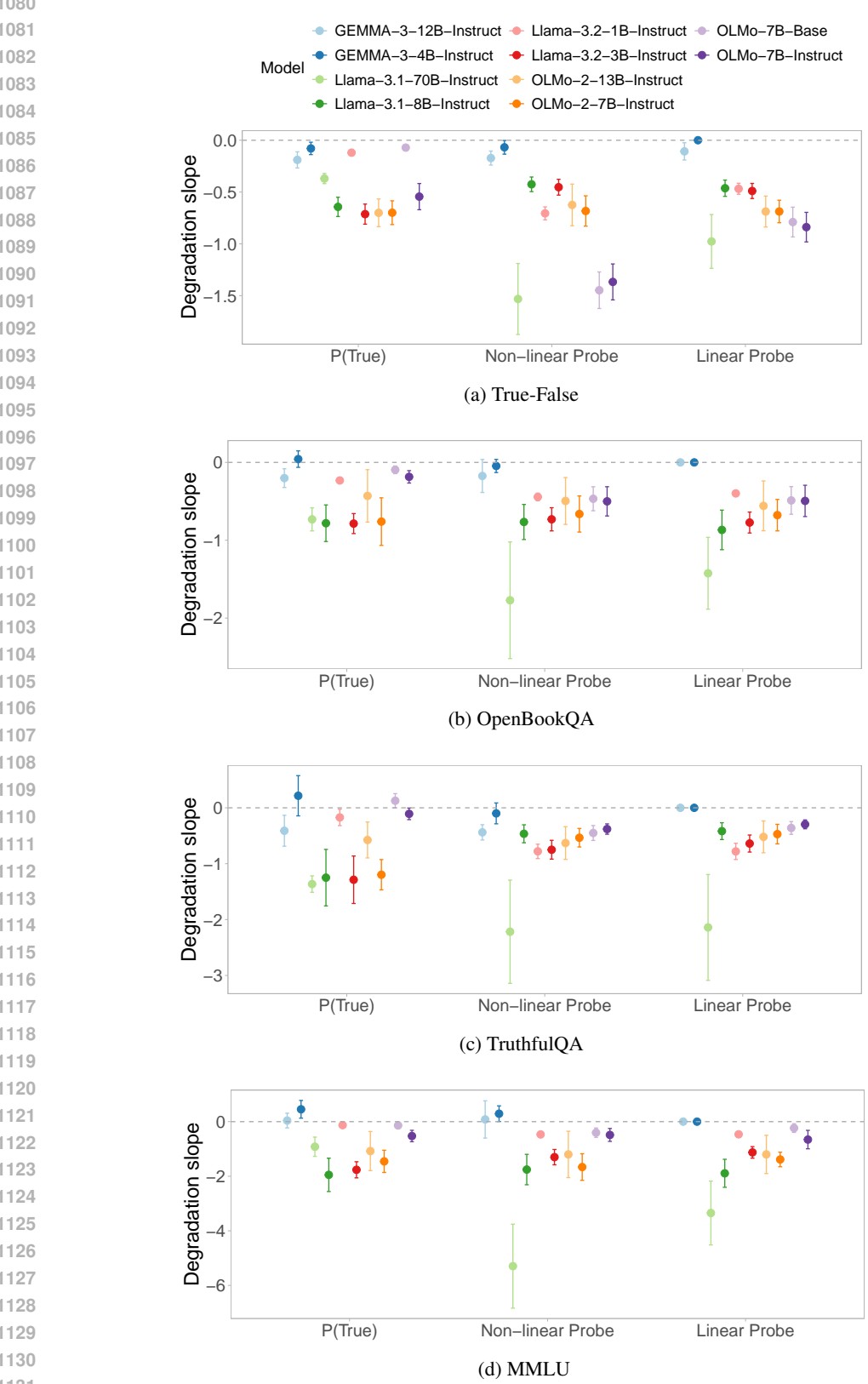

Fig. S3: Magnitudes (mean ± standard error) of degradation slopes for all probing methods and model combinations across the four different datasets.

tive, giving a false impression of robustness. Thus, it is impossible to compare the robustness of this smaller model against the larger models, so it is excluded from our analysis.

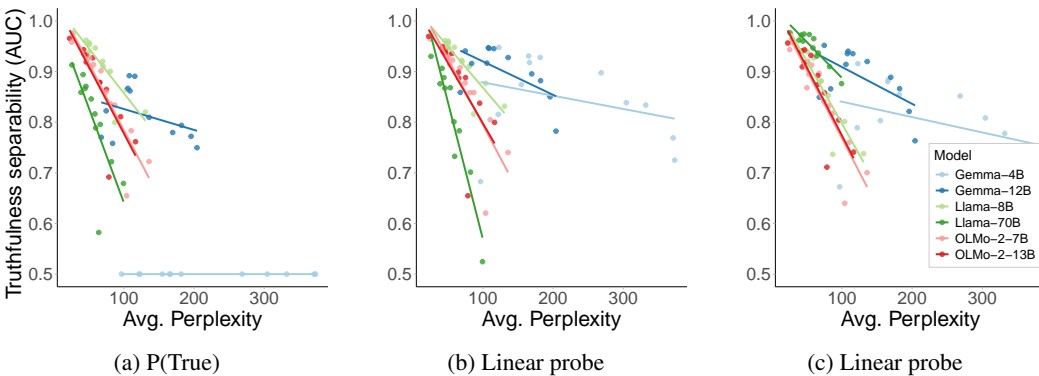

(a) P(True)       (b) Linear probe       (c) Linear probe

Fig. S4: **(a)** linear probe, **(b)** non-linear probe, and **(c)** P(true) performance (AUC) against average perplexity for various model families on True-False dataset.

## C.4   INTER-LAYER ANALYSIS OF REPRESENTATION ROBUSTNESS

To address the concern that truthfulness degradation might occur in layers other than the best-performing in terms of AUC on the original, un-transformed dataset, we conducted an inter-layer analysis on a representative model–dataset pair.

**Setup.** We re-ran the non-linear probe experiments on `Llama 3.1 8B` using the True-False dataset. For this analysis, we extracted hidden representations from all even-numbered layers (2, 4, 6, ..., 32). For each layer, we trained and evaluated a probe under increasing OOD shifts and plotted truthfulness separability (AUC) as a function of average statement perplexity.

**Results.** We show the degradation curves for all layers in Figure S5 and in tabular format in Table S5. We observe three consistent patterns across layers:

(1) **Lower layers** begin near chance-level AUC on the original data and remain near chance across all OOD conditions. (2) **Middle layers**, which typically correspond to the best-performing layers reported in our main results, show high separability in-distribution and clear degradation under OOD shift. (3) **Upper layers**, including the final layer, also exhibit high separability on untransformed samples but show the steepest degradation as inputs become more OOD.

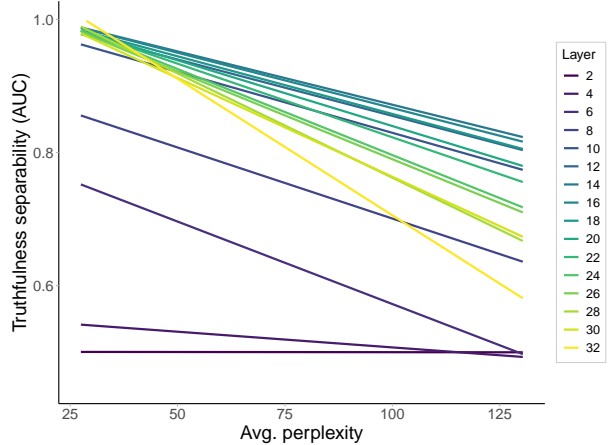

Fig. S5: AUC vs. perplexity across layers of Llama 3.1 8B Instruct on True–False. **Middle and upper layers degrade under OOD shift, while lower layers remain at chance.**

**Conclusion.** This analysis shows that any layer that encodes truthfulness in-distribution—middle and upper layers alike—exhibits consistent collapse as statements move out-of-distribution. Lower layers, which do not encode truthfulness to begin with, remain near chance regardless of OOD shift. Therefore, reporting results from the "best-performing" layer in the main paper is a conservative choice: even the most informative layers demonstrate the brittleness we highlight, and the inter-layer trends reinforce our central conclusion.

| Layer | Initial AUC | Degradation Slope |
|-------|-------------|-------------------|
| 2 | 0.500 | -0.001 |
| 4 | 0.571 | -0.083 |
| 6 | 0.813 | -0.436 |
| 8 | 0.901 | -0.376 |
| 10 | 0.961 | -0.322 |
| 12 | 0.971 | -0.298 |
| 14 | 0.979 | -0.282 |
| 16 | 0.976 | -0.293 |
| 18 | 0.975 | -0.305 |
| 20 | 0.971 | -0.348 |
| 22 | 0.970 | -0.389 |
| 24 | 0.960 | -0.455 |
| 26 | 0.962 | -0.463 |
| 28 | 0.960 | -0.551 |
| 30 | 0.960 | -0.522 |
| 32 | 0.955 | -0.722 |

Table S5: Initial AUC and degradation slopes for each layer of Llama-3.1-8B on the true-false dataset using the non-linear probe.

## C.5 CERTAIN TOPICS HAVE MORE ROBUST REPRESENTATIONS

In this appendix, we provide topic-level analyses complementing the main results in §5. Specifically, we include figures illustrating relationships between benchmark accuracy, statement length, and robustness, and the full set of topic-wise statistics for the non-linear probe on Llama 3.1 8B.

Figure S6 are included to further contextualize the by-topic results from §5. Fig. S6a illustrates the relationship between LM Evaluation Harness accuracy and the non-linear probe AUC, showing that overall, high benchmark scores are associated with high separability of true from false statements. In Fig. S6b, we show that while there is no clear association between average (by-topic) statement lengths and degradation slopes, we note that topics with particularly large average statement length exhibit higher absolute degradation rates (i.e., lower robustness).

In Table S6, we report the full by-topic results for the non-linear probe based on Llama 3.1 8B, presented in Fig. 6, including degradation slopes and probe AUC as well as additional metrics such as benchmark accuracy, average perplexity, log-average n-gram count, and average sentence length.

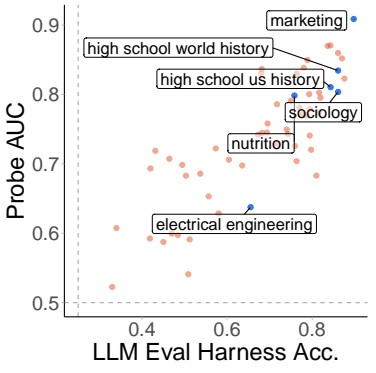
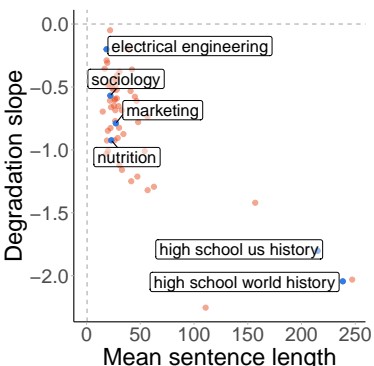

(a) Benchmark accuracy vs. probe AUC.  (b) Mean sentence length vs. degradation.

Fig. S6: **(a)** Benchmark accuracy vs. non-linear probe AUC and **(b)** mean sentence length vs. degradation rate of non-linear probe based on Llama 3.1 8B Instruct. **Topics with high benchmarks tend to exhibit high true-false separability.**

## C.6 TYPE OF TRANSFORMATION

Figure S7 shows the change in AUC of each transformed True-False dataset variant against the change in average perplexity for all three probes. We note similar trends for all three probing methods, except in the case of negation. For both the linear and non-linear probes, negation leaves average perplexity unchanged and has no effect on AUC, indicating that truthfulness representations remain stable under syntactic negation. By contrast, P(True), depicted in Fig. S7c, exhibits a noticeable drop in AUC despite no corresponding change in perplexity, suggesting that while negated statements are not driven out-of-distribution, the output-based method is nonetheless sensitive to the transformation and yields lower separability.

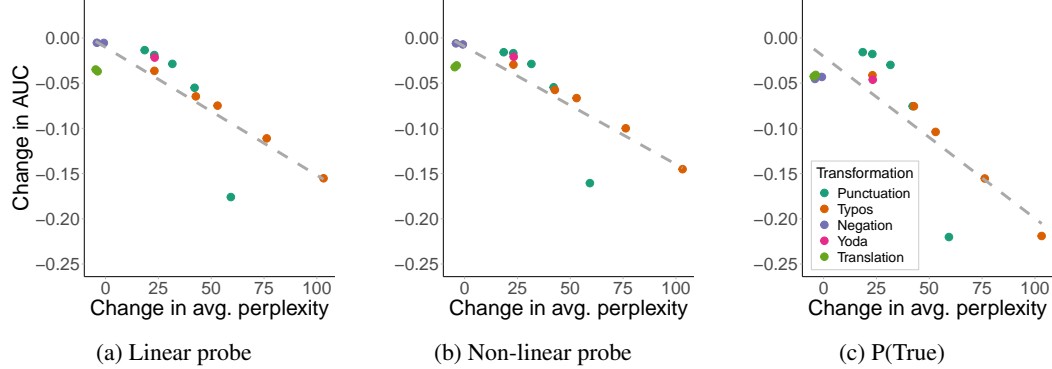

(a) Linear probe         (b) Non-linear probe         (c) P(True)

Fig. S7: Change in **(a)** linear probe, **(b)** non-linear probe, and **(c)** P(true) performance (AUC) against change in average perplexity on True-False dataset.

# D REPRODUCIBILITY

Below we provide code samples for the non-linear and linear probe methods to aid reproducibility. For P(True), see Kadavath et al. (2022).

## D.1 NON-LINEAR PROBE

```python
class MLPClassifier(nn.Module):

    def __init__(
        self,
        input_dim: int = 4096,
        hidden_dims: list = [256, 128, 64],
    ):
        super().__init__()

        layers = []
        in_dim = input_dim
        for h_dim in hidden_dims:
            layers.append(nn.Linear(in_dim, h_dim))
            layers.append(nn.ReLU())
            in_dim = h_dim

        layers.append(nn.Linear(in_dim, 1))

        layers.append(nn.Sigmoid())

        self.net = nn.Sequential(*layers)

    def forward(self, x: torch.Tensor) -> torch.Tensor:
```

```
          return self.net(x).squeeze(-1)
```

## D.2 LINEAR PROBE

```python
class LogisticRegressionClassifier(nn.Module):

    def __init__(self, input_dim: int = 4096):
        super().__init__()
        self.linear = nn.Linear(input_dim, 1)
        self.sigmoid = nn.Sigmoid()

        self.net = nn.Sequential([
            self.linear,
            self.sigmoid
        ])

    def forward(self, x: torch.Tensor) -> torch.Tensor:
        return self.net(x).squeeze(-1)
```

Table S6: Per-topic statistics: slope, AUC, accuracy, perplexity, n-gram count, and sentence length for MMLU topics (Llama-3.1-8B-Instruct, non-linear probe).

| Topic | Slope | AUC | Eval. acc | Perp. | n-gram c. | sent. len |
|---|---|---|---|---|---|---|
| abstract algebra | -0.44 | 0.61 | 0.34 | 8.93 | 5.09 | 27.6 |
| anatomy | -1.01 | 0.84 | 0.68 | 10.10 | 7.59 | 20.0 |
| astronomy | -0.65 | 0.70 | 0.76 | 10.85 | 5.76 | 25.6 |
| business ethics | -0.65 | 0.83 | 0.68 | 17.09 | 6.15 | 29.8 |
| clinical knowledge | -0.93 | 0.85 | 0.79 | 11.53 | 7.46 | 19.0 |
| college biology | -1.12 | 0.80 | 0.82 | 9.98 | 6.99 | 30.6 |
| college chemistry | -0.52 | 0.60 | 0.47 | 8.56 | 7.54 | 28.1 |
| college computer science | -0.74 | 0.63 | 0.58 | 8.10 | 7.81 | 56.8 |
| college mathematics | -0.19 | 0.52 | 0.33 | 6.53 | 6.80 | 38.4 |
| college medicine | -1.01 | 0.75 | 0.69 | 10.34 | 7.33 | 54.2 |
| college physics | -1.25 | 0.72 | 0.43 | 6.61 | 7.56 | 41.6 |
| computer security | -0.41 | 0.78 | 0.77 | 20.03 | 4.55 | 28.3 |
| conceptual physics | -0.29 | 0.71 | 0.60 | 16.65 | 4.14 | 18.7 |
| econometrics | -0.36 | 0.54 | 0.51 | 9.59 | 7.16 | 42.2 |
| electrical engineering | -0.20 | 0.64 | 0.66 | 19.29 | 4.27 | 18.6 |
| elementary mathematics | -0.69 | 0.70 | 0.49 | 9.28 | 6.89 | 26.8 |
| formal logic | -0.58 | 0.60 | 0.48 | 12.27 | 3.44 | 45.0 |
| global facts | -0.51 | 0.69 | 0.42 | 10.09 | 12.74 | 23.5 |
| high school biology | -1.16 | 0.80 | 0.82 | 9.35 | 7.20 | 33.0 |
| high school chemistry | -0.82 | 0.70 | 0.64 | 8.26 | 6.99 | 30.3 |
| high school computer science | -1.21 | 0.75 | 0.74 | 8.33 | 7.30 | 47.3 |
| high school european history | -2.03 | 0.83 | 0.76 | 8.26 | 6.62 | 247.2 |
| high school geography | -0.48 | 0.80 | 0.79 | 15.18 | 6.23 | 19.7 |
| high school government and politics | -0.92 | 0.82 | 0.88 | 11.40 | 7.78 | 26.1 |
| high school macroeconomics | -0.52 | 0.74 | 0.68 | 14.52 | 6.39 | 25.0 |
| high school mathematics | -0.38 | 0.59 | 0.42 | 6.72 | 7.24 | 30.6 |
| high school microeconomics | -0.77 | 0.74 | 0.79 | 12.22 | 6.52 | 26.6 |
| high school physics | -0.61 | 0.59 | 0.45 | 6.10 | 6.95 | 46.6 |
| high school psychology | -0.91 | 0.86 | 0.86 | 12.62 | 6.55 | 29.0 |
| high school statistics | -1.32 | 0.69 | 0.54 | 6.68 | 8.33 | 56.8 |
| high school us history | -1.80 | 0.81 | 0.84 | 6.34 | 7.84 | 215.0 |
| high school world history | -2.04 | 0.83 | 0.86 | 8.50 | 7.12 | 238.4 |
| human aging | -0.31 | 0.76 | 0.70 | 18.83 | 4.95 | 19.3 |
| human sexuality | -0.66 | 0.78 | 0.79 | 14.54 | 6.10 | 22.8 |
| international law | -0.59 | 0.68 | 0.81 | 12.01 | 6.03 | 28.2 |
| jurisprudence | -0.59 | 0.77 | 0.78 | 14.53 | 5.49 | 26.5 |
| logical fallacies | -0.60 | 0.72 | 0.80 | 14.70 | 5.69 | 26.1 |
| machine learning | -0.87 | 0.71 | 0.46 | 12.66 | 6.38 | 34.4 |
| management | -0.36 | 0.77 | 0.82 | 17.38 | 4.45 | 16.9 |
| marketing | -0.79 | 0.91 | 0.90 | 15.44 | 6.17 | 27.3 |
| medical genetics | -1.03 | 0.84 | 0.78 | 11.58 | 7.30 | 18.6 |
| miscellaneous | -0.85 | 0.87 | 0.84 | 13.33 | 6.07 | 19.8 |
| moral disputes | -0.44 | 0.74 | 0.74 | 21.04 | 5.23 | 25.6 |
| moral scenarios | -1.29 | 0.72 | 0.57 | 13.27 | 11.79 | 62.7 |
| nutrition | -0.92 | 0.80 | 0.76 | 11.25 | 7.86 | 23.1 |
| philosophy | -0.49 | 0.79 | 0.72 | 18.83 | 4.20 | 21.6 |
| prehistory | -0.83 | 0.79 | 0.75 | 14.04 | 6.58 | 22.3 |
| professional accounting | -0.78 | 0.65 | 0.56 | 9.48 | 8.69 | 48.0 |
| professional law | -1.42 | 0.68 | 0.50 | 6.13 | 8.82 | 157.0 |
| professional medicine | -2.25 | 0.77 | 0.79 | 4.99 | 10.15 | 110.8 |
| professional psychology | -0.69 | 0.73 | 0.72 | 13.22 | 6.34 | 32.9 |
| public relations | -0.49 | 0.74 | 0.67 | 15.96 | 5.89 | 26.9 |
| security studies | -0.53 | 0.73 | 0.76 | 13.69 | 7.10 | 41.0 |
| sociology | -0.57 | 0.80 | 0.86 | 16.03 | 6.47 | 22.2 |

| Topic | Slope | AUC | Eval. acc | Perp. | n-gram c. | sent. len |
|---|---|---|---|---|---|---|
| us foreign policy | -0.61 | 0.85 | 0.87 | 14.11 | 5.78 | 21.8 |
| virology | -0.05 | 0.59 | 0.51 | 15.16 | 6.19 | 21.9 |
| world religions | -0.70 | 0.87 | 0.84 | 15.80 | 4.86 | 15.1 |

