# OpenReview forum: "LLM knowledge is brittle: truthfulness representations rely on superficial resemblance"
_ICLR.cc/2026/Conference — Submitted to ICLR 2026_

### Official Review · Reviewer_k7BS · 2025-10-28

**Soundness:** 3
**Presentation:** 3
**Contribution:** 2
**Rating:** 4
**Confidence:** 4

**Summary:**

This paper examines the brittleness of knowledge representations in Large Language Models (LLMs) by testing how well they maintain truthfulness separability under out-of-distribution (OOD) transformations (e.g., typos, Yoda-speak, translations). Using perplexity as an OOD proxy, the study applies linear, non-linear, and P(True) probes across four LLM families and datasets (True-False, MMLU, OpenBookQA, TruthfulQA), finding that separability degrades with increasing OOD-ness, suggesting reliance on superficial training data resemblance.

**Strengths:**

The study is comprehensive, testing across multiple LLM families (10 models total), datasets, probing techniques, and transformation types. This broad scope enhances the generalizability of findings within the evaluated setups.

By linking OOD-ness (via perplexity) directly to representation separability, the paper offers a plausible mechanistic explanation for why LLMs struggle with subtle input variations.

Clear Visualization: The paper is well-written and easy to follow, with figures (e.g., AUC vs. perplexity plots) effectively illustrating degradation trends.

**Weaknesses:**

As noted, the core finding—that OOD data leads to worse probe performance—aligns with prior work on truthfulness probes (e.g., [1] cited by the authors). Degraded separability on perturbed data has been observed in similar contexts. Thus, this paper does not introduce novelty on ideas or methods.

Also, the P(True) experiments also show degraded performance, consistent with expectations based on prior output-based brittleness studies, suggesting the findings are not unexpected.

Lack of Deep Interpretability: The degraded separability is still the observation. Observing degraded separability does not provide deeper insights into LLM knowledge brittleness. The paper attributes this to "shallow" representations but lacks exploration.

Overall, this paper finds the correlation of degraded representation with data OOD. However, this is under expectation, and this paper lacks further explanation.

[1] Levinstein et al.Still no lie detector for language models: Probing empirical and conceptual roadblocks.

**Questions:**

See weakness part.

---

> ### Author Response · Authors · 2025-11-21
> **Response to Reviewer k7BS (1/2)**
>
> We thank the reviewer for their thoughtful and constructive assessment of our paper. We appreciate the positive remarks regarding the breadth of our evaluation across multiple model families, datasets, and probing methods, as well as the clarity of our figures and writing.
>
> **Novelty of results**
>
> We appreciate the reviewer’s concern about novelty and the connection to prior work showing degraded probe performance under certain transformations. We would like to clarify how our contribution differs in scope and substance from existing studies, including Levinstein et al.
>
> **Prior work evaluates probe generalization, not representation robustness.**
> Much of the existing literature, including Levinstein et al., shows that probes trained on in-distribution data fail to generalize to specific perturbations such as negation. These works highlight limitations of *probing as a methodology*, rather than limitations of the *underlying LLM representations*.
>
> In contrast, our design avoids the probe-generalization confound entirely: **we train and test probes within each OOD-shifted distribution** (i.e., the probe is trained and tested on the transformed data).
>
> This ensures the probe has direct access to the transformed representations. When separability still collapses under these conditions, this provides evidence that the problem lies in the **representations themselves**, rather than in probe generalization. While previous work has, we agree, pointed out limitations of probe generalization, our work goes further by isolating a deeper, more fundamental problem with the robustness of learned representations themselves. The finding of non-robust knowledge representations has far broader implications for robust and reliable LLMs than that of non-generalizable probes.
>
> **Systematic link to distribution shift.**
> Specifically, our work is the first to systematically link degree of out-of-distribution shift (via perplexity) to failure of truthfulness separability. Prior work has shown degradation under isolated transformations (e.g., negation), but to the best of our knowledge, no prior study has established a continuous relationship between (a) how far a sample is pushed OOD with respect to the model’s own distribution, and (b) how sharply truthfulness separability collapses. This OOD–separability linkage is the central novelty of our paper and reveals a mechanistic explanation for brittleness that goes beyond documenting probe failures.
>
> **Breadth of evaluation.**
> Our findings extend across models, datasets, and probe types, including output-based methods. Whereas earlier studies focus on single models, limited transformations, or single datasets, our results hold across:
>
> * 10 models from 4 families,
> * 4 diverse benchmarks, and
> * 3 probing approaches.
>
> In summary: Robust and reliable LLMs should be able to learn generalizable knowledge representations that can be deployed in a wide range of novel contexts, including those that are unlike the training data (i.e., OOD). Our work provides evidence that LLMs do not learn such generalizable representations, providing a potential explanation for brittle benchmark performance.

---

> ### Author Response · Authors · 2025-11-21
> **Response to Reviewer k7BS (2/2)**
>
> **Lack of Deep Interpretability**
>
> While our core empirical finding is indeed that truthfulness separability degrades under OOD shift, our contribution goes further than that. We agree that deeper insight into *why* representations degrade is essential, and we aimed to provide this through several analyses that investigate what factors predict representation brittleness. In particular, the paper offers multiple layers of interpretability beyond the headline result:
>
> * **Dataset-, model-, and transformation-level comparisons:** By systematically varying datasets, model families, and transformation types, we identify which forms of distribution shift are most disruptive. For example, Fig. 7 shows that certain shifts (e.g., translation) break separability even when they do not increase perplexity, suggesting that syntactic reorganization affects model representations differently than surface-level noise.
> * **Topic-level analysis revealing variation across knowledge domains:** In Fig. 6a-c, we analyze robustness by fine-grained MMLU topics. This reveals that (1) some topics (e.g., sociology, marketing) are consistently more robust, others (e.g., world history) degrade sharply, and (2) topic-level robustness is not explained by pre-training data coverage. This provides insight into which kinds of knowledge are learned more or less robustly, and why brittleness is not uniform across domains.
> * **Cross-probe consistency:** Because we observe the same patterns across linear probes, nonlinear probes, and the output-based P(True) method, the degradation cannot be attributed to a probing artifact. This supports the interpretation that the brittleness originates in the representations themselves.
>
> Taken together, these analyses provide a deeper understanding of how and when LLM knowledge representations fail, and identify properties of both the input and the underlying distribution that govern this failure. We will clarify these interpretability contributions more explicitly in the revised paper.
>
> Once again, we thank the reviewer for their comments, and hope our clarifications about our contribution—including how our experimental setup and key results significantly differ from existing work on probe generalization—are useful. We are of course happy to discuss further and answer any remaining questions.

---

### Official Review · Reviewer_DeBU · 2025-10-31

**Soundness:** 3
**Presentation:** 2
**Contribution:** 2
**Rating:** 4
**Confidence:** 4

**Summary:**

This paper investigates whether the widely observed brittleness of LLM performance stems from unstable internal knowledge representations. The authors apply semantically-preserving transformations (typos, punctuation noise, syntactic reordering, and translation) to drive evaluation samples out-of-distribution, then measure how well trained probes can separate true from false statements using internal representations. Across 10 models from 4 families (OLMo, Llama, Gemma), 4 datasets (True-False, MMLU, OpenBookQA, TruthfulQA), and 3 probing methods (linear probes, non-linear probes, and P(True)), the results consistently show that truthfulness separability degrades as samples become more out-of-distribution, as measured by perplexity. Fine-grained analyses reveal that certain topics (e.g., sociology, marketing) are more robust than others (e.g., history), though this variation is not explained by pre-training data coverage. Notably, representation robustness shows little correlation with whether models answer questions correctly in standard benchmarking settings, suggesting a disconnect between task performance and internal knowledge encoding.

**Strengths:**

1. This paper addresses an important and timely research question: whether the widely observed brittleness in LLM benchmark performance stems from unstable internal knowledge representations or merely from task execution issues. By attempting to distinguish between "brittle performance" and "brittle knowledge", the authors provide a new perspective on understanding LLM capabilities and limitations. The findings, if validated, could have implications for both pre-training strategies and post-training alignment methods.

2. The paper demonstrates commendable experimental breadth, evaluating 10 models across 4 model families, 4 diverse datasets, and 3 distinct probing methodologies. This comprehensive evaluation across multiple dimensions, spanning different model architectures, knowledge domains, and probing techniques, strengthens the generalizability of the observed patterns.

**Weaknesses:**

1. The authors claim that the applied transformations (typos, Yoda-style reordering, and translation) are semantically preserving. However, translated sentences or those with Yoda-like word order may not be strictly semantically equivalent and could disrupt the syntactic dependencies of the original content. The paper does not include any human validation or semantic-similarity evaluation to confirm the equivalence of the transformed samples. This ambiguity undermines the interpretation of the observed "degradation": does it truly indicate that the model's knowledge representations are non-robust, or does it simply reflect natural activation shifts caused by syntactic restructuring?

2. The experimental setup is insufficiently described, and the probe's training data remain unspecified. Although the related-work section states that "our probes are both trained and tested in out-of-distribution settings", lines 828–829 claim that the probes were "trained for 5 epochs on a balanced dataset of true/false". It is unclear whether transformed OOD samples were used for probe training, under the paper's current description, these would all correspond to false samples, making the setup highly ambiguous.

3. In practice, the method reduces to using the hidden states of the entire [question + candidate answer] pair as encoder representations, combined with externally provided true/false labels, to train a binary classifier as a probe. These labels do not reflect the LLMs' own beliefs, meaning the LLM is effectively treated as a frozen "encoder". In a classification pipeline, we would rarely attribute non-robustness to the encoder when the downstream classifier underperforms because the issue could simply stem from the probe itself. Therefore, the causal inference drawn in the paper, i.e., probe performance drops on OOD data → representation non-robust (representation collapse), is questionable. It is equally plausible that the degradation originates from the probe's limited generalization capacity. While the authors acknowledge the limitations of probing, I remain unconvinced by the claim that the LLM's internal representations are brittle. Even though multiple probes yield consistent results, most are linear or shallow MLPs, which are too simplistic to support such a strong conclusion.

4. The authors report results from the "best-performing layer" but provide no analysis of inter-layer variation. If representational differences across layers are substantial, the observed "degradation" might occur only in the upper decoding-related layers rather than in the core knowledge layers. Selecting the single best layer could exaggerate or obscure the true behavior of the model’s internal representations.

**Questions:**

1. Were the probes trained solely on the original (in-distribution) data, or was a separate probe trained for each dataset? In Figures 2–5, what do the plotted points for different datasets represent, are they generated by the same probe evaluated under different OOD transformations, or by distinct probes trained per dataset? Specifically, was each probe trained on the "Original" data and then tested on the transformed OOD variants, or were the probes re-trained for each OOD condition?

2. **Please address the questions in the Weakness section.**

---

> ### Author Response · Authors · 2025-11-21
> **Response to Reviewer DeBU (1/2)**
>
> We thank the reviewer for their careful assessment of our work and for the thoughtful comments provided. We appreciate the recognition of the importance of distinguishing between brittle performance and brittle knowledge, as well as the acknowledgment of the breadth and generality of our experimental evaluation across models, datasets, and probing methods. We will address the reviewer’s noted weaknesses in detail below.
>
> **1. Human validation**
>
> We appreciate the reviewer’s suggestion of a human validation. To address this, we are conducting a small-scale human study. The goal of this evaluation is to confirm that our transformations do not alter the truth value for humans, and therefore that the observed degradation reflects brittleness in model representations, not changes in human-perceived semantics. We will report back with results as soon as they are ready.
>
> **2. Description of experimental setup**
>
> We thank the reviewer for pointing out the ambiguity in our description of the probe training procedure. We will revise the methods section accordingly to clarify this point and will share an updated draft when ready. We would like to clarify three aspects relating to label balance, probe training, and the rationale for our approach:
>
> * **Label balance:** Transformed (OOD-shifted) samples do not correspond to “false” statements, and we never relabeled them as such. Every transformed sample retains the same ground-truth truth value as its original version. For example, if “Berlin is a city in Germany” is transformed via typos or translation, all variants remain labeled as true. Thus, our balanced true/false datasets remain balanced under all transformation conditions.
> * **Probe training procedure:** For every dataset and every transformation condition, the probes were always trained on the transformed (OOD-shifted) samples themselves—not only evaluated on them. That is, each probe was trained and tested within the same transformation setting, using stratified cross-validation, such that the probe is a direct test of the separability of the true/false statements when all statements have undergone the OOD shift.
> * **Rationale:** Our goal is to measure the robustness of the model’s own internal representations. Therefore, probes must be trained on the transformed representations themselves to give them the best chance to learn the correct mapping between representation and truth value. If truthfulness separability collapses even when trained directly on transformed samples, this provides strong evidence that the representations themselves—not the classifier—have become non-robust.

---

> ### Author Response · Authors · 2025-11-21
> **Reviewer DeBU (2/2)**
>
> **3. Strong conclusions**
>
> We thank the reviewer for this thoughtful critique. We agree that distinguishing between probe robustness and representation robustness is crucial. However, we believe that there is a misunderstanding regarding our experimental setup that addresses this specific concern:
>
> 1.  The reviewer suggests that the performance drop might stem from the probe’s limited generalization capacity. This would be a viable interpretation if we trained the probe on original data and tested it on OOD data. However, this is not our methodology. As detailed in Section 3 (Methods), we train and test probes *within* the out-of-distribution datasets. More formally, let $\mathcal{D}_ o$ be the original dataset, and let $\mathcal{D}_ {t}$ be the transformed (OOD) dataset. The reviewer’s concern assumes a setup of:
>     $ \text{Train}(\mathcal{D}_ {o}) \rightarrow \text{Test}(\mathcal{D}_ {t}) $
>     (i.e., training the probe on original data and testing the probe on transformed data). In this scenario, a drop in performance could indeed be attributed to the probe failing to generalize to the new distribution.
>
>     Our actual setup is:
>     $\text{Train}(\mathcal{D}_ {t}) \rightarrow \text{Test}(\mathcal{D}_ {t})$
>     (i.e., training the probe to separate representations of transformed data, and testing on a test split of that same distribution). Because the probe is trained directly on the representations of the transformed samples, it does not need to generalize from the original distribution. Consequently, if the probe fails to classify true/false statements in $\mathcal{D}_ {t}$ (despite having high capacity and working perfectly on $\mathcal{D}_ {o}$), it indicates that the representations themselves have become inseparable.
>
> 2.  The reviewer argues that in a pipeline, we would "rarely attribute non-robustness to the encoder." Were the performance drop attributable to the probe, we would agree. However, given the frozen encoder produces features that are linearly (and non-linearly) separable for $\mathcal{D}_ {o}$, but, crucially, inseparable for $\mathcal{D}_ {t}$, we instead conclude that the encoder is **brittle** with respect to the transformation used to create $\mathcal{D}_ {t}$. The encoder has failed to map semantically equivalent inputs (e.g., a statement with a typo) to a similar, separable region in the latent space.
>
> 3.  Regarding the labels: while the True/False labels are externally provided, the high separability we observe on untransformed data confirms that the LLM’s internal representations do encode the information necessary to align with these labels. The fact that this separability disappears under superficial perturbation—even when we train a new probe to find it—supports our claim that the truthfulness representation is unstable.
>
> 4.  Finally, regarding the simplicity of probes: We employ both linear and non-linear probes, and the output-based P(True). We observe that even non-linear probes trained on transformed data fail to recover separability (even when an equivalent non-linear probe can recover separability on the original data). This suggests that the degradation is not merely a result of the decision boundary becoming non-linear or the probe lacking capacity.
>
> **4. Inter-layer analysis**
>
> We thank the reviewer for raising this point. Our decision to report results from the best-performing layer follows common practice in probing work (e.g., Azaria & Mitchell, 2023), but we agree that examining inter-layer variation can provide additional insight. To address this concern, we are now working on an additional analysis of inter-layer variation. We will report back with results as they are ready.

---

> ### Author Response · Authors · 2025-12-03
> **Results of inter-layer analysis**
>
> As promised in our initial response, we conducted the additional inter-layer analysis to address the reviewer’s concern that the observed degradation might occur only in upper decoding-related layers.
>
> We re-ran the non-linear probe on **Llama 3.1 8B** using the **True–False dataset**, extracting activations from all even-numbered layers (2, 4, 6, …, 32). For each layer, we plotted truthfulness separability (AUC) against statement perplexity.
>
> **Findings:**
>
> - **Lower layers** start near chance AUC on the original samples and remain near chance across OOD shifts, indicating that these layers do not encode truthfulness.
> - **Middle layers** — typically those selected as “best-performing” in the original analysis — show strong separability in-distribution and clear degradation under OOD shift.
> - **Upper layers**, including the final layer, also exhibit high in-distribution separability but show the steepest degradation as inputs become more OOD.
>
> **Conclusion:**
> This analysis demonstrates that representation degradation is not only observable in upper decoding layers. Any layer that encodes truthfulness (middle and upper layers) shows consistent collapse under OOD shift, with the most pronounced effects in the top layers. Thus, our use of the “best layer” in the main paper is conservative and does not exaggerate the observed brittleness.
>
> We will add this analysis and the corresponding figure to the Appendix.

---

### Official Review · Reviewer_Vfu7 · 2025-11-01

**Soundness:** 3
**Presentation:** 3
**Contribution:** 3
**Rating:** 2
**Confidence:** 5

**Summary:**

The paper investigates the brittleness of internal knowledge representations in large language models (LLMs) by examining how well truthfulness is encoded and how robust this encoding is to out-of-distribution (OOD) inputs. Using a range of semantically-preserving transformations the authors test truthfulness separability across multiple LLMs, datasets, and probing methods. The study reveals that truthfulness representations degrade significantly as inputs become less similar to pre-training data, demonstrating that LLMs rely heavily on superficial surface-form similarities. This brittleness persists across models and datasets, raising concerns about the robustness of knowledge encoded by LLMs and their reliability in real-world applications.

**Strengths:**

- Comprehensive benchmark design – The evaluation framework is thorough, incorporating multiple model architectures, probing methods and diverse datasets.
- Clear conceptual framing – The distinction between “brittle performance” and “brittle knowledge” is valuable and helps situate the paper within the broader robustness literature.
- Methodological clarity – The probing setup and transformations are well-documented, ensuring reproducibility and transparency.

**Weaknesses:**

- Validity of transformations - some of the sentence transformations are unlikely to occur naturally, raising questions about how the findings translate to everyday use cases. Some natural paraphrasing or longer contextual shifts could improve ecological validity.

- No human validation – There is no human-based validation of how meaningful the observed “truthfulness degradation” is, or whether humans perceive similar brittleness.

- Limited discussion of solutions – While the results clearly expose brittleness, the discussion of practical steps to improve robustness is limited

- Scope of claim – The discussion occasionally implies that all knowledge representations are brittle, though the experiments focus only on truthfulness separability. Clarifying this boundary would improve precision.

**Questions:**

Please see the concerns raised above.

---

> ### Author Response · Authors · 2025-11-21
> **Response to Reviewer Vfu7**
>
> We thank the reviewer for their thoughtful and constructive feedback. We are grateful for the positive assessment of our benchmark design, as well as the clarity of the conceptual framing and methodology. We will address the reviewer’s noted weaknesses in detail below.
>
> **Validity of transformations**
>
> We thank the reviewer for raising this important point. A key clarification is that our definition of “out-of-distribution” (OOD) is explicitly with respect to the model’s own learned representations of the training data, not with respect to user behavior. While our chosen transformations are selected because they shift samples out-of-distribution, many of the transformations, such as typos and punctuation noise, are in fact common in real user inputs.
>
> Our results show that any OOD shift, regardless of how naturalistic from an everyday use case, causes systematic degradation in truthfulness representations. This has direct implications: Even naturally occurring variations (e.g., typos) can produce unreliable internal representations if they differ sufficiently from the model’s training distribution.
>
> Regarding natural paraphrases, we agree that including more naturalistic transformations would be a beneficial addition. However, having explored this during the project’s implementation, we found that automatically generating paraphrases that (1) strictly preserve truth value and (2) meaningfully shift distribution is challenging. Most paraphrases remain close to the model’s pre-training distribution and therefore do not induce OOD shifts.
>
> If the reviewer were able to provide further clarity on what they have in mind for “longer contextual shifts”, we’d be happy to consider this.
>
> **Human validation**
>
> We thank the reviewer for highlighting the need for human validation. In response to this helpful feedback, we are now running a human validation study in which annotators judge whether each transformed statement has the same truth value as its original form.
>
> This directly tests whether the transformations meaningfully change the content for humans. Our aim is to show that the transformations do not alter truth values for humans—i.e., humans do not suffer the brittleness we expose for models. We will report back with results as soon as completed.
>
> **Discussion of solutions**
>
> We thank the reviewer for highlighting the importance of discussing potential solutions to improve robustness. Our goal in this paper is to identify and characterize a problem—namely, that current LLMs exhibit non-robust internal knowledge representations—rather than to propose a recipe for eliminating this brittleness. We agree, however, that outlining promising solution directions is valuable, and we will expand the discussion section accordingly.
>
> While our experiments suggest that simply scaling model size or increasing pre-training data coverage does not reliably improve topic-level robustness, we see several exciting avenues for future work. These include:
>
> * **Fine-tuning on OOD-shifted examples** to investigate whether targeted exposure can stabilize internal representations,
> * **Exploring data augmentation during pre-training** (e.g., injecting controlled perturbations or linguistic variability) as a way to encourage invariances, and
> * **Modifying the pre-training objective** to explicitly encourage robustness to surface-form variation.
>
> **Scope of claim**
>
> We thank the reviewer for raising this important distinction regarding the scope of our claims. We consider truthfulness separability to be a kind of knowledge test: being able to assign an accurate truth value requires accurate knowledge.
>
> We therefore argue that truthfulness separability serves as a fundamental baseline: because it is a binary task, it is arguably less complex than open-ended generation or multi-step reasoning. Consequently, the brittleness we observe in this setup suggests that our estimates are rather conservative. Robustness may degrade even more severely in complex tasks that require more fine-grained aspects of knowledge representations. Testing brittleness in these settings remains an exciting area for future research.
>
> Once again, thank you for your helpful feedback and we remain available should you have further questions.

---

> > ### Author Response · Authors · 2025-12-03
> > **Results of human validation**
> >
> > As committed in our initial response, we conducted a human validation study to determine whether our transformations (Yoda-order, translation, typos) preserve truth value for humans.
> >
> > **Study design:**
> >
> > - 90 statements, each with an original and a transformed version (plus original–original pairs as a baseline).
> > - Three transformations: **Yoda-transform**, **punctuation noise**, **typos**.
> > - Four annotators, each seeing only one version of each statement.
> > - A Latin square design ensured that across annotators, both original and transformed versions of each item were evaluated exactly once.
> >
> > Annotators were asked to assign a binary truth value (true/false) to the version they saw. For each item, we compared the truth-value assignments between annotators who saw original versions and those who saw transformed versions.
> >
> > **Results:**
> > Across all transformation types, the inter-annotator agreement between original–transformed pairs fell between **0.778 and 0.800**, which is effectively identical to the agreement observed for original–original pairs.
> >
> > **Conclusion:**
> > These results show that the transformations are semantically preserving for humans. If transformations altered truth value or semantics, we would expect substantially lower agreement between original–transformed annotations compared to the original–original baseline. Instead, humans maintain stable truth-value judgments regardless of transform.
> >
> > We will incorporate these results and a description of the study into the Appendix.

---

### Meta-Review · Area_Chair_3EZZ · 2026-01-16

**Summary:**

All reviewers agreed the paper is well executed and clearly written, but looking at the discussions, the overall suggest a weak reject because several core concerns remain. Reviewers questioned whether some “semantically preserving” transformations (especially Yoda-style reordering and translation) truly preserve meaning, and noted that without strong validation the observed degradation could reflect syntactic/input effects rather than brittle knowledge. They also flagged ambiguity in the probe training/setup in the original submission, which undermined confidence until clarified. More fundamentally, reviewers were not fully convinced that lower probe AUC under OOD conditions uniquely implies representational brittleness, since it may still reflect limits of the probe/readout or extractability. Finally, one reviewer argued the main result is not sufficiently novel relative to prior work showing probe brittleness under perturbations (one could lower the weight on the lack of novelty here. Indeed, the paper follow similar type of work but  I would subjectively state that the specific attempt to diagnose representation brittleness vs probe brittleness by training and testing probes within each shifted distribution plus a continuous OOD metric is a clearer experimental angle than many earlier papers), and another noted the paper sometimes overreaches in scope, implying broad “knowledge brittleness” while only testing truthfulness separability, with limited discussion of realistic settings and mitigations. Perhaps some overclaims could be tempered or at least strongly demonstrated in the numerical experiments since no clear theoretical explanations if offered.

**Reviewer Concerns:**

The rebuttal resolves several clarity and validation issues: it explains that probes are trained and tested within each transformed setting (labels are preserved), adds a human study supporting that the tested transformations preserve truth value for humans, and adds an inter-layer analysis showing the degradation appears in layers that encode truthfulness. Still outstanding are concerns about lack of natural paraphrases/longer context shifts, some remaining uncertainty around translation, and the main debate about whether lower probe separability under OOD proves “brittle knowledge” versus a probe/readout or input-processing effect; novelty concerns are also only partly addressed but this is a bit harder to settle.

**Reviewer Scores:**

- Reviewer Vfu7: likely stays 2, rebuttal helps with human validation, but most concerns likely remain and keep a reject score.
- Reviewer DeBU  likely be on weak accept (setup clarified + inter-layer + human validation reduce key concerns, but causal attribution may still worry them).
- Reviewer k7BS likely stays 4 (main issues are novelty and depth of insight, which the rebuttal mostly reframes rather than substantively changes). This could be harder to move, even thought with more discussions, the authors could revisit the litterature review and more clearly distinguish the novelty of the paper

---

### Decision · Program_Chairs · 2026-01-26

Reject